# In the Shadow of the Casinos: The Relationship between Religion and Health in Macau

**DOI:** 10.3390/ijerph19095605

**Published:** 2022-05-05

**Authors:** Yiyi Chen, Jiaqi Lu, Canghai Guan, Shiyang Zhang, Spencer De Li

**Affiliations:** 1Department of Sociology, University of Macau, Macao 999078, China; chenyiyi1062@126.com (Y.C.); jqlu1118@hotmail.com (J.L.); yb97320@umac.mo (C.G.); zhangshiyang@bnu.edu.cn (S.Z.); 2School of Future Design, Beijing Normal University, Zhuhai 519085, China

**Keywords:** religion, altruism, prejudice, life satisfaction, health, Macau

## Abstract

Considerable research has shown that religion operates as a protective factor for one’s health. However, there is still a lack of understanding of the mechanisms by which religion is linked to individual health and wellbeing, especially in predominantly secular societies. This study tried to address this gap by developing a theoretical model to examine how religiosity is related to life satisfaction and health perception in a non-Western culture. Macau, a Portuguese colony until 1999, remains a diversified culture because of its intermixed historical background from the East and the West. Through structural equation modeling, the analysis of data collected from a representative sample of Macau residents, using a multistage stratified sampling procedure, indicated a positive link between religiosity and health. Moreover, altruism and prejudice mediated a portion of the relationship between religiosity and health. Additionally, our results demonstrated that Macau residents who were more religious had a higher level of altruism and a lower level of prejudice. The link between religion and prejudice in Macau differs from that of many other cultures, indicating that the effect of religion on prejudice varies by cultural context. In sum, our study showed that even in the shadow of glittering casinos, religion is positively related to health.

## 1. Introduction

The link between religion and health has received increasing attention in recent decades. Most of the research has indicated that religious commitment serves as a protective factor for one’s overall health. Religion is an important component of many people’s lives, and it is represented by individuals’ beliefs and practices that make them feel connected to God or Sacred figures [1]. By health, we mean “a state of complete physical, mental and social wellbeing” that is defined by the World Health Organization [2]. Some studies have discovered a positive relationship between religiosity and body functioning [3,4], self-rated health [5,6,7], life satisfaction [8,9], hope [10,11], optimism [12,13], and positive traits [14,15,16,17]. Religion has also been found to be negatively related to depression [18,19,20], anxiety [21,22], suicide [23,24], and substance abuse [25,26]. Furthermore, religious involvement is considered to provide “a favorable impact on a host of physical diseases and the response of those diseases to treatment” [27] (p. 9). This statement is supported by a host of studies showing that religion is negatively related to the presence of coronary heart disease [28,29], hypertension [30,31], cerebrovascular disease [32], Alzheimer’s disease or dementia [33,34], and cancer [35,36]. For instance, research that correlated religion with cancer treatments indicates that the provision of religious support to cancer patients can improve their recovery and reduce their malign symptoms [35]. The importance of religion to health is acknowledged in the suggestive adoption of religious practices in health care systems [37,38], and many countries have emphasized the integration of religious and spiritual aspects into nursing and health treatments.

In fact, much of the prior research about the relationship between religion and health is undertaken in the continents of the world that contain the greatest proportion of religious believers. The understanding of religiosity and health is largely based on the evidence from western cultures, where a large segment of the population holds strong religious identities and beliefs. For example, in the United States (U.S.), about 65% of Americans consider religion an important part of their life; over 90% of them believe there is God or a higher being; 83% of them pray to God each week, and 43% of them report having attended religious services almost every week [39,40]. Some studies about the relationship between religion and health in the U.S. show that infrequent religious attenders are more likely than frequent attenders to have impaired physical functioning, chronic health conditions, and fair or poor self-rated health [5,41]. Furthermore, people with consistent religious service attendance display greater health outcomes and lower mortality rates [40,42].

Comparable research in Europe has also explored the positive correlation between attending religious activities and health [43]. These studies add to the findings above, which present that individuals who have more religious service participation will have lower odds of chronic disease attacks, fewer functional impairments, higher health ratings, fewer hospitalizations, and better coping with diseases than those with lower religious service attendance [44,45,46,47,48]. A few related studies indicate that the practice of religion by any individuals with or without illnesses is beneficial because it provides them with a greater sense of mission, broader mindset, greater self-care, healthier lifestyles, and more positive emotional support for coping with unexpected health problems [40,49,50,51]. 

Conversely, in non-Western settings, like China, a limited number of studies have examined the relationship between religion and health. Examples of these studies are based on small or religiously homogenous samples, which undermines the generalizability of the findings. Macau, located in the south of the Pearl River Delta area, is a special administrative region (SAR) of China, and it is recognized by many as the “Las Vegas of the East” because it arguably holds the world’s largest commercial gambling operation [52,53]. Indeed, Macau is a secular society that relies on commercial gambling for its government revenue and personal income. For people living in Macau, personal wealth and material success are highly valued due to the casino culture of reward-seeking and auspiciousness from wins and losses. 

Alongside Macau’s casino culture, religiosity exists in the lives of many Macau residents, even though it is a predominantly secular society. Macau was a former Portuguese colony that embraced both Western and Eastern traditions. Besides the religions originating from the East, like Buddhism, Daoism, and Mazuism, Macau also contains many followers of religions originating from the West, such as Catholicism and Protestantism. However, research on the relationship between religion and health among Macau residents is very limited because there are few studies relevant to the current topic. One of them describes the relationship between religious beliefs and sleep disturbance [54], while the second examines the mediating effect of religious belief and religious school attendance on death anxieties among adolescents [55]. According to prior surveys, at least 30% of Macau residents identify themselves as religious, and over 35.7% of them report attending religious activities at least once a week [56]. Additionally, Macau has a relatively high life expectancy of 84 years old [57], and its secular nature, along with its religious diversity, makes it an explorable context for delving into the relationship between religion and health. Therefore, the purpose of this study is trying to narrow the gap in the understanding of religion on overall health outcomes among Macau residents. 

## 2. Theoretical Background

Religiosity may not be directly related to health outcomes. However, religious beliefs and devotion have been presumed to indirectly influence individual health through intermediary psychological and behavioral pathways [27]. Many psychosocial studies have confirmed that religion works as a “meaning system” in believers’ lives [58,59], that is, the important resource of sense-making that can shape individuals’ beliefs and attitudes. Most religions preach specific value orientations, such as conformity, generosity, and benevolence [60,61]. Religious participation provides individuals with an interface to these values and beliefs. The “meaning system” of religion protects individuals’ health by not only helping them cope with adversities, setbacks, and negative emotions [62] but also the promotion of meaning and adoption of normative values [63]. Additionally, the effect of religiosity is more substantial among elderly people. For example, many elderly people are considered healthy because they derive a sense of meaning and purpose in life from religion [64]. Holding religious beliefs cultivate intangible support for their emotional or socio-economic conditions, and the devotion to God and religious participation further alleviate their sense of helplessness [58]. 

Henceforth, we expect religiosity to improve one’s health through enhancing positive psychosocial traits such as altruism and neutralizing negative attitudes such as prejudice. Adopting religious values of generosity, benevolence, and helping the poor and needy is commonly referred to as being altruistic. Because of the greater appreciation of other people, it is uncommon for religious believers to attribute others negatively, assuming a lesser likelihood of prejudicing others. Thus, this study expects that religion can promote altruism and neutralize prejudice, both of which serve as intermediary factors for people’s health. 

### 2.1. Altruism as a Mediating Mechanism for Religion and Health

Altruism is a prosocial behavior performed by someone with a strong motivation to help others, usually under a sense of empathy and perspective-taking, and research shows that altruism and health can be positively correlated [16,65]. Altruism can be distinguished into formal and informal helping behaviors, with the former involving contributing unpaid time to formal organizations and the latter involving helping individuals or contributing to informal organizations [66]. Altruism from formal helping behavior is found to be associated with longer life expectancy, better self-rated health, and improved emotions [67,68]. Informal helping behaviors, such as peer help between patients or community members, can reduce depression and increase self-confidence and self-esteem, all positively affecting health [69]. Meanwhile, any type of formal or informal helping behavior can increase individuals’ life satisfaction and happiness [70,71]. 

Religious activities tend to provide their believers with greater altruism, in which they offer assistance unconditionally and do good voluntarily for others [72,73]. Most research results indicate that individuals with a religious affiliation are more willing to help strangers than those without a religious affiliation. Altruism also comes from an increased sense of benevolence and generosity in religious believers who frequently participate in religious activities that revolve around their religious beliefs [49,61]. The benevolence and generosity of religious individuals cultivate their authentic concerns for other people and influence their motivation to help others [74]. In addition, some studies show that individuals who possess a greater inclination to help others report a sense of divine control in which God sends them on the mission of being of aid to others [75]. The commitment to helping others is found to be strongest in places where the enforcement of religiosity is a matter of personal choice [76]. When religion is a personal choice rather than socially imposed, individuals can foster their autonomy through internalizing moral and social values that originate from their intrinsically motivated deeds rather than external ones. 

Because of the prevailing role of the gambling industry in Macau, it is expectable that “casino capitalism” may propagate undesirable behaviors and beliefs that evolve collective interest into individual pursuits, such as the egocentrism commonly found in the capitalistic ethos [77,78]. Thus, we expect that religion may help compress individualistic pursuits through accreting altruistic mindsets. That is, people with greater religious commitment should demonstrate a higher level of altruism. Given the positive relationship between religion and altruism from prior research, and the positive association between altruism and health, it is justifiable to assume that altruism can mediate the link between religion and health.

### 2.2. Prejudice as a Mediating Mechanism for Religion and Health

According to prior literature, religion can either potentially promote or mitigate an individual’s prejudice against others [79,80]. Nevertheless, studies that suggest a positive relationship between religion and prejudice are more prevalent in countries with a more individualistic culture. In other words, countries that emphasize collectivism with less pronounced individual power will display an opposing relationship between religiosity and prejudice [81]. For instance, religious individuals in regions of Japan, Korea, mainland China, and Taiwan that are more collectivistic in culture tend to have lower prejudice toward out-groups [82,83]. There is also evidence that one’s religious view will influence their level of prejudice. Believers with a stronger commitment to the religious ideas of “benevolence”, “morality”, and “interpersonal guilt” tend to be more caring, selfless, and sympathetic and will avoid doing things that might harm others [84]. 

Furthermore, religious individuals can be divided into intrinsically religion-motivated and extrinsically religion-motivated, with the former internalizing religious and social values and the latter being more utilitarian in their religious and social values [85]. According to Jackson and Hunsberger, intrinsically religion-motivated believers show less prejudice toward people who do not hold the same beliefs as them compared to extrinsically religion-motivated believers because they tend to be more neighbor-loving, compassionate, and humble [86]. Thus, people who attend more religious services will demonstrate less prejudice if they have greater internalized religious and moral values, even in individualistic cultures.

Many studies show that prejudice can be negatively related to one’s health. Besides harming the targets of prejudice, prejudice against others is also found to be harmful to discriminators’ physical and psychological health [87,88]. According to the previous literature, people with less prejudice predict lower anxious mood, less physiological threat relating to cardiovascular reactivity, increased levels of anabolic hormones, and a lower mortality rate [89,90,91,92]. These findings suggest that egalitarian attitudes toward cross-social and cross-ethnic groups can provide physical and psychological benefits to people living in culturally diversified societies.

As over 95 percent of its population is composed of ethnic Chinese, Macau is deeply rooted in the Chinese cultural tradition and normative system. Like Taiwan and Hong Kong, it can be considered largely a collectivistic society where religious commitment is more internally motivated as opposed to externally motivated. As such, we expect that religiously committed individuals in this society possess less prejudice against social and ethnic groups that are different from their own. Considering the evidence that lower prejudice may promote mental and physical health, it is reasonable to expect prejudice to mediate the relationship between religiosity and health in Macau. 

### 2.3. Religion, Life Satisfaction, and Health

The concept of life satisfaction philosophically and psychologically accounts for both individuals’ happiness and well-being [93], and it is regularly identified as a significant component of one’s health. To be more specific, the meaning of life satisfaction is a hedonic state derived from a judgment that involves the comparison of individuals’ actual level of pleasure with their desired level of pleasure. The positive relationship between life satisfaction and overall health has been extensively documented. Individuals who rate their life satisfaction higher are considered to be healthier [94], be more physically fit [95], have higher longevity [96], be less susceptible to chronic illnesses [97], and have fewer lingering illnesses [98].

Research that explored religion as an influence on health indicates that participation in religious activities and religious salience are significantly related to life satisfaction. For example, attendance in religious services can promote individuals’ mental well-being as well as their physical health. Specifically, religion enhances individuals’ health through provision of greater content with life, which reflects higher satisfaction with life [99]. A study focusing on group variation in religious affiliations finds that religious believers who are Buddhists, Protestants, and Catholics report greater life satisfaction than non-religious individuals. There is also an inference that participation in religious activities enables believers to hold more optimistic attitudes toward their life and adopt healthier behaviors [100,101] since individuals with greater life satisfaction are linked to more positive subjective feelings and better emotional reactions under the presence of stressors [102]. As such, enhancement of life satisfaction is partially due to the autonomy of choice and emotional expression provided by religions to their believers, and all of this improves the believers’ health outcomes [103]. This suggests that greater life satisfaction, exhibited by religious involvement, can promote a person’s health by pushing away negative effects that often increase one’s susceptibility to illnesses [104].

## 3. Current Study

Despite prior evidence that supports religion being a protective factor for one’s health, few studies have systematically explored the relationship between religion and health in predominantly secular societies. Thus, the purpose of this study is to advance our understanding of the relationship between religiosity and health in a non-Western culture. Besides the direct relationship between religion and health, this study will examine several indirect mechanisms by which religion might promote individual health. Based on the review of theories and the prior literature, it is conceivable that religion may be related to health through the mediating effects of altruism or prejudice, or both. To be more specific, this study will try to address the following questions: (a) how might religiosity be related to individual health status? (b) does religiosity increase one’s altruism? (c) does religiosity decrease one’s prejudice against others? (d) does altruism mediate the relationship between religiosity and life satisfaction? (e) does prejudice mediate the relationship between religiosity and life satisfaction? (f) does altruism mediate the relationship between religiosity and overall health? (g) does prejudice mediate the relationship between religiosity and overall health?

The current study hypothesizes that higher religiosity can lead to more altruism, which in turn increases satisfaction with life and overall health. Moreover, higher religiosity is hypothesized to decrease prejudice against others, thereby alleviating its negative influence on life satisfaction and overall health. In sum, religiosity is expected to be directly related to life satisfaction and health and indirectly related to life satisfaction and health through its positive relationship with altruism and negative relationship with prejudice. Figure 1 below illustrates the theoretical relationships between the key concepts that are described above. 

## 4. Methods

### 4.1. Sample and Data

The present study employed the data collected in the Macau Social Survey (MSS), a project funded by the University of Macau and conducted by the research team at the university. The Panel on Research Ethics of the university reviewed and approved the study design on the 15 October 2015 (project number: MYRG2015-00116-FSS). A stratified cluster systematic probability proportional to size sampling method was used in this study to ensure the representativeness of the sample. According to the census conducted by the Statistics and Census Service (DESC) of Macau SAR, the region consists of 23 land statistical areas, excluding dwellings that are off the coast, to serve as the stratification factors. Based on the geographic information provided by the DESC, a list of housing units, e.g., residential buildings and other types of dwelling units, was constructed for each stratum. In the project, a household was defined as a group of individuals who share living expenses and reside in the same accommodation unit that’s being surveyed. Eligible participants are residents who were at least 16 years old and have stayed in Macau for 3 consecutive months in the past 6 months, counting from the time of the interview. 

After sampling was completed, trained interviewers distributed an invitation letter to each sample household, informing them of the survey’s purpose and assuring them of the privacy and confidentiality that the survey complies with. Any household that declined the invitation to participate in the study was replaced by another sampling household randomly drawn from the same stratum. Upon obtaining approval, the researchers then visited the household and conducted interviews with the eligible subjects. The household survey was conducted through Computer-Assisted Personal Interviewing that contained a structured questionnaire covering a broad range of social issues important to Macau society. The questions included but were not limited to employment, marriage and family, norms and attitudes, political and civic participation, volunteering, religion, and health. Modeled after the General Social Survey and European Social Survey [105,106], this is the largest population-based survey ever conducted in Macau, focusing on a wide range of social indicators and their correlates.

The current study is based on an analytic sample of 3258 respondents, representing 93% of the total sample of 3502 individuals from 2601 households. Applying the listwise deletion procedure, 244 participants are excluded from the analyses because of missing responses on some key variables.

### 4.2. Variables and Measures

To the extent possible, we used standard instruments to measure the key concepts in our theoretical model. The measures of religiosity and health perceptions were based on the General Social Survey, while the questions measuring altruism were selected from the Human Value scale in the European Social Survey. Other instruments adopted in this study are listed below when applicable. To minimize measurement errors, we used multiple indicators to measure the theoretical constructs. The measure of prejudice was preconstructed, while the other factors were latent variables measured using multiple indicators. For these latent variables, confirmatory factor analysis was performed to assess how well the indicators represent each of the corresponding theoretical constructs. All the estimated factor coefficients of the indicators were larger than 0.65 and significant at the *p* < 0.001 level, indicating that they loaded on the corresponding construct (See Appendix A for further information). The following descriptions list all the variables and measures included in our analytical model. 

*Religiosity.* Religiosity is the extent to which an individual is committed to the religious teachings that are not only represented by his/her attitudes but also the behaviors that reflect this commitment. Church attendance is frequently used to measure religiosity in previous studies [107,108]. However, there have been concerns about the validity of using church attendance as a single indicator of religiosity [109,110]. As suggested in past research, the concept of religiosity has both attitudinal and behavioral dimensions [111,112,113]. Thus, we operationalized the latent concept of religiosity using three indicators: (1) frequency of attending religious activities (henceforth “religious activity attendance”), (2) importance of religion to the respondent (“religious salience”, hereafter), and (3) the respondent’s belief in God. The question measuring the first dimension asked how often the respondent attended religious activities. The response options ranged from “0 = never attended” to “7 = attended several times a week”. The second question asked the respondent to rate the importance of religion to him or her on a scale from “1 = not important” to “5 = very important”. The third question asked the respondent’s belief in God from “1 = I don’t believe in God” to “6 = I know God really exists and I have no doubts about it”. The Cronbach’s alpha computed from the three questions is 0.75, indicating good reliability. Factor loadings of the three questions on the construct of religiosity were 0.79, 0.74, and 0.70, respectively.

*Altruism.* In past research, the most frequent measures of altruism included caring for friends and families, communities and societies, and the world [114]. Similarly, our measures of altruism tapped into three dimensions: helping people, making contributions to society, and caring for nature. The respondents were asked to indicate how well they think each statement represents them by checking one of the response options ranging from “1 = not like me at all” to “7 = exactly like me”. The questions included: “It’s very important to him/her to help the people around him/her”; “Making positive contributions to the society is important to him/her”; “He/she strongly believes that people should care for nature. Looking after the environment is important to him/her”. The respondents’ altruism, thus, was inferred from their self-rated scores on the statements resembling themselves. The Cronbach’s alpha calculated from these three items is 0.78, indicating good reliability. Factor loadings of the three variables on the construct were 0.66, 0.86, and 0.69, respectively.

*Prejudice.* Prejudice refers to a hostile attitude or feeling toward an individual solely because the person is perceived as belonging to a group in which one has assigned objectionable qualities [115]. In this study, we measured prejudice using the social distance scale, which has been adopted by prior research as a measure for this concept [116]. In the interview, the respondent was asked to check any of the groups of individuals listed on the survey questionnaire that they were unwilling to have as next-door neighbors. The choices included drug users, people from different racial groups, patients diagnosed with AIDS, immigrants, foreign laborers, homosexuals, people of different religions, alcoholics, unmarried cohabitating couples, and people speaking a different language. The respondent received a value of 1 for any group they checked and 0 otherwise. The sum of all their responses concerning the nominated groups constituted the measure of prejudice.

*Life Satisfaction.* Life satisfaction was measured by a 5-item instrument, developed by Diener and his colleagues [117], that has been used to assess individuals’ satisfaction with their lives on a cognitive level with good psychometric properties and high test-retest reliability. The questions included, “In most way my life is close to my ideal.”; “The conditions of my life are excellent.”; “I am satisfied with my life.”; “So far I have gotten the important things I want in life.”; “If I could live my life over, I would change almost nothing”. All items were rated on a 7-point scale with “1 = strongly disagree” up to “7 = strongly agree”. The value of Cronbach’s alpha derived from these items is 0.93, suggesting a very high level of reliability. The factor loading of the five items on the construct were 0.83, 0.89, 0.90, 0.90, and 0.72, respectively.

*Health.* For health, the respondents were asked to compare their own health condition to that of their peers, to their health from last year, and to self-assess their general health. The questions on peer comparison were answered based on a five-point scale ranging from “1 = much worse than peers” to “5 = much better than peers”. Similarly, the response categories for cross-year comparison ranged from “1 = much worse than last year” to “5 = much better than last year”. For self-assessed general health, respondents reported their health condition by rating from “1 = my health is poor” to “5 = my health is excellent”. The value of Cronbach’s alpha computed from the items is 0.83, indicating a very high level of reliability. Factor loading of the three questions on the construct of overall health were 0.79, 0.76, and 0.85, respectively.

*Problem Gambling.* Considering the prevalence of commercial gambling in Macau, the analysis included problem gambling to control the potential impact of this addictive behavior on health. This variable was measured by a 9-item instrument developed for assessing the severity of problem gambling in social surveys, the Problem Gambling Severity Index (PGSI) [118]. All items on the frequency of at-risk gambling behaviors use a 4-point scale with “0 = Never” to “3 = Almost always”. The scores of each item were summed up, resulting in a composite measure ranging from 0 to 27. Following the scoring instruction, the gambling risk levels were then categorized into “0 = no-problem”, “1 = low-risk”, “2 = moderate-risk”, and “3 = problem gamblers”. The value of Cronbach’s alpha derived from these items is 0.91, suggesting a very high level of reliability. 

This test of the theoretical model also included age, gender, birthplace, and education level as the control variables in the analysis. Age was an interval variable measured by a number that represents the time elapsed since the date of the respondents’ year of birth. Considering that religious commitment and health fluctuate across the lifespan, controlling for age increases the reliability of estimates of the impact of religiosity on health. Gender was a dichotomous variable, with “1” representing “female” and “0” representing “male”. Birthplace was also a dichotomous variable, with “1” for Macau and “0” for “anywhere outside Macau”. The education level was measured by asking the respondents to rate their highest level of education, and it consisted of categories ranging from “0 = never received any education” to “7 = doctoral degree”.

## 5. Results

### 5.1. Characteristics of the Sample

Table 1 presents the distribution of all study variables that describe the basic characteristics of the sample. Around 52.30 percent of respondents were female, with the average age being about 43.78 years old. About 1709 respondents in the sample were born in Macau, accounting for 52.46 percent of the total sample. About 44.05 percent of respondents had completed less than a middle school education; 22.59 percent of respondents had completed high school education; 33.36 percent of them had received an education level of college or higher. The mean rating for problem gambling was 0.06, suggesting that addictive behaviors in problem gambling are relatively rare among Macau residents.

Among the non-demographic variables, the mean scores for the three dimensions of religiosity were 1.06, 2.15, and 3.09, respectively, representing average scores of religious activity attendance, religious salience, and belief in God. Second, the mean ratings for altruism and life satisfaction were all greater than 4.55, indicating relatively high levels of altruism and satisfaction with life. Third, prejudice had a mean score of 1.77, indicating a relatively low level of prejudiced attitude against others. Finally, the respondents scored mean values of 3.28, 3.07, and 2.69 on the three measures of health on a 5-point scale.

Table 2 details the distribution of religious affiliations. As shown in the table, around 31.62 percent of respondents reported being affiliated with at least one religious organization. The most prevalent religion in Macau was Buddhism, which approximately accounted for 15.56 percent of the total respondents. About 6.51 percent of respondents considered themselves Catholic, and 4.73 percent of respondents considered themselves Protestant. The fourth religious category, “other”, representing about 3.87 percent of the total respondents, includes traditional folk religions as well as other lesser-known religions from Asia, such as Hinduism. Additionally, believers in Daoism, Mazuism, and Islam accounted for 0.68 percent, 0.15 percent, and 0.15 percent of the total respondents.

### 5.2. Results of Structural Equation Models

SEM analysis was performed to test the structural model to examine how religiosity was related to respondents’ satisfaction with life and overall health. The results of the test are illustrated in Figure 2. Religiosity was directly and positively related to life satisfaction (β = 0.038). The relationship, however, was not statistically significant. Apart from the direct effect, religiosity was related to life satisfaction through the mediating effects of altruism and prejudice. To be more specific, religiosity positively predicted altruism (β = 0.126, *p* < 0.001), and negatively predicted prejudice (β = 0.089, *p* < 0.001). Thus, respondents with a higher level of religiosity tended to have a greater degree of altruism and a lower degree of prejudice. In addition, the level of altruism was positively related to life satisfaction, with a standardized coefficient of 0.247 (*p* < 0.001); prejudice was negatively linked to life satisfaction, with a standardized coefficient of −0.157 (*p* < 0.001). 

In terms of the influence of religiosity on overall health, the direct effect was significant and positive (β = 0.181, *p* < 0.001). This result is consistent with the findings of prior research indicating that religiosity serves as a protective factor for individual health. In addition to its direct effect, religiosity also predicted health through altruism and prejudice. Altruism was found to be positively related to overall health (β = 0.097, *p* < 0.001), that is, respondents with a higher degree of altruism tended to have a more favorable perception of their overall health. Prejudice was negatively related to overall health, with a standardized coefficient of −0.206 (*p* < 0.001); that is, less prejudiced respondents had a better rating of their health. Presented also in Figure 2 is the positive relationship between life satisfaction and overall health (β = 0.152, *p* < 0.001), where respondents with a higher level of life satisfaction appeared to have better self-reported health. All these results have lent support to the hypothetical relationships proposed in Figure 1. These findings held true even after controlling for the variables of age, gender, birthplace, education level, and problematic gambling behaviors. 

Altruism and prejudice, according to the findings, partially mediated the effect of religiosity on life satisfaction as well as overall health. Their direct, indirect, and total effects are further decomposed in Table 3. The total effect between religiosity and life satisfaction was significant and positive, with a standardized coefficient of 0.083 (*p* < 0.001). The direct effect of religiosity on life satisfaction accounted for approximately 45.78 percent of the total effect, while 54.22 percent of the total effect was explained by indirect effects through altruism (β = 0.031, *p* < 0.001) and prejudice (β = 0.014, *p* < 0.001). 

With regards to the relationship between religiosity and overall health, the total effect was found to be significant and positive (β = 0.225, *p* < 0.001). Along with the direct effect discussed above, all the mediators, except for life satisfaction, significantly contributed to an overall positive relationship between religiosity and health. The indirect pathway from religiosity to overall health could be explained by different mediating mechanisms. More specifically, religiosity demonstrated a significant indirect effect on overall health through the mediating mechanisms of altruism (β = 0.012, *p* < 0.01) and prejudice (β = 0.018, *p* < 0.001). That is, religiosity enhanced altruism and mitigated prejudice, which in turn improved the individuals’ health. Similar results could be found in the indirect pathways from religiosity to overall health through life satisfaction. The mediating mechanisms took place in the following ways: through higher altruism, religiosity established a positive relationship with life satisfaction; then, through enhanced life satisfaction, religiosity attained a positive and significant relationship to health (β = 0.005, *p* < 0.01). Simultaneously, through lower prejudice, religiosity assumed a positive relationship with life satisfaction and then through increased life satisfaction, religiosity significantly improved overall health (β = 0.002, *p* < 0.01). 

In summary, religiosity is shown to improve an individual’s overall health directly and indirectly through fostering altruism and reducing prejudice. The indirect effect of religiosity on health explained around 20% of the association between religiosity and health in the current study, while the rest constitutes the direct effect unmediated by any variable included in the model.

## 6. Discussion

The main objective of this study is to assess the relationship between religion and health in Macau, a special administrative region of China with the largest commercial gambling market in the world [119]. Because of the confluence of Eastern and Western traditions, Macau has the most diverse population in China in terms of religious commitment and participation. Over 30 percent of the population aged 16 and above identified themselves as affiliated with at least one religion, with Buddhism, Catholicism, and Protestantism being the most frequently affiliated religions. While this percentage is high in the Greater China Region consisting of mainland China, Taiwan, Hong Kong, and Macau, it is considerably lower than many other regions in the world. For instance, about 75% of adults possess a religious faith in the U.S [120]. Compared to 36 percent in the United States and 72 percent in Indonesia for religious participation [121], roughly 11 percent of the respondents in our survey reported attending religious activities at least once a week. Therefore, Macau is a predominantly secular society with great cultural and religious diversity, making it a unique place to study the potential influences of religion on health.

Prior research has documented interrelationships among religiosity, life satisfaction, and health, but there is more to be learned about the mechanisms underlying these relationships. Thus, to provide a better understanding of the ways in which religion might influence health, we proposed a theoretical model incorporating both direct and indirect effects of religiosity on health through altruism and prejudice. The findings obtained from the analysis of the data collected in the Macau Social Survey largely supported our hypotheses. Our study results showed that religiosity was positively related to altruism and negatively related to prejudice among the residents in Macau. Moreover, those who had stronger altruistic attitudes and less prejudice also reported higher levels of satisfaction with life and overall health. Hence, altruism and prejudice operated as important mechanisms linking religiosity to health. While altruism and prejudice mediated much of the relationship between religiosity and satisfaction with life, they showed a lesser contribution to the link between religiosity and health, suggesting that religiosity might promote health conditions through other forms that were not accounted for in this study. Overall, consistent with our theoretical expectations, religiosity could contribute to individual health in multiple ways. 

The mediating role of prejudice is noteworthy. In many predominantly religious societies, such as the Islamic societies in the Middle East and selected regions of the United States, religious followers tend to uphold more conservative values, including a higher level of prejudice against non-religious people, individuals who maintain unconventional moral beliefs, or groups who engage in different lifestyles involving drug use and homosexuality [122,123]. The opposite seems to be true in Macau. Our results demonstrated that Macau residents who were more religious had a lower level of prejudice. This finding is consistent with the prior research showing that the minority status of religious believers fosters the acceptance of socially disadvantaged groups in Macau [56]. As a minority living in a society dominated by consumerism from the gambling industry, they felt that their ways of life and their value systems were increasingly marginalized, which led to more sympathy and acceptance for other minority groups who might have also experienced social or cultural marginalization. Moreover, the attempt to bring into harmony religious beliefs and moral values to neutralize the experience of social marginalization indicates that most Macau residents tend to be intrinsically religious believers. It is perhaps because of these reasons that religion serves to curtail prejudice and thereby contribute to overall health among Macau residents. Thus, the link between religion and prejudice in varying contexts and its effect on health are worthy of further exploration.

Much of the knowledge about the relationship between religion and health is based on studies conducted in the societies where a large portion of the population is affiliated with only a few dominant religions. It is unclear if the results obtained from these societies apply to other societies that are either predominantly secular or more religiously diverse. Macau fits both descriptions because nearly 70 percent of its population self-identify as non-religious, with the rest considering themselves as religiously affiliated with many different kinds of Eastern and Western religions. Despite the secular context and the religious diversity, religion still contributed significantly to health among the residents in Macau. These findings are in line with the results of several other studies showing that religion might have a salutary effect on health outcomes in secular cultures. For instance, a longitudinal study of a Danish cohort born in 1914 found a negative correlation between church attendance and mortality among these elders in the secular culture [124]. A correlational study conducted in Denmark indicated that women who reported lower religiosity had poorer self-rated health and higher illness severity [125], whereas men displayed an inverse relationship between religious practice and health. The discrepancy is due to the fact that Danish women were more religiously attached than men, indicating that intrinsically religious individuals were more likely to benefit from religious commitment in terms of health outcomes [126]. Other research conducted in secular cultures also demonstrated that individuals who participated in religious activities developed more positive meaning-making systems, healthier dietary patterns, and lesser involvement in risky health behaviors, which could all contribute to improved health conditions [127,128]. These overlaps between the research findings from Macau and those from other regions lend credence to the proposition that religion can promote individual health in cultures that are secular in nature.

Despite the contributions that it has made to the understanding of the relationship between religion and health, the results of the current study should be taken with caution. First, the research design was cross-sectional. Therefore, the relationships identified in the analysis are correlational in nature and should be interpreted as such. Second, the measures of overall health used in this study were based on the respondents’ perceptions through self-reported data. Although self-reporting represents a reasonable way to measure health, given our focus on overall health as a state of physical, mental, and social well-being [2], it may be biased by respondents’ interpretation of their own health status, especially when it needs to be compared with others. Third, confirmation bias could have played a role in influencing the respondents’ responses to the survey questions on prejudice and altruism, as some of them might have interpreted the questions in a way that partially reflected their existing beliefs or expectations [129]. Future research should employ a longitudinal design with an improved ability to draw a causal inference based on empirical findings to address these limitations. Furthermore, they should consider incorporating reports of multi-informants such as doctors, friends, and family members to improve the reliability of the health measures.

## 7. Conclusions

The present study developed a theoretical model examining the relationship between religiosity and overall health in a predominately secular society. Consistent with the findings from Western societies, our results showed that religiosity was significantly and positively related to satisfaction with life and overall health among the residents in Macau. That is, people with a higher level of religiosity were more satisfied with their lives and reported better health. Religiosity appeared to contribute directly to overall health. Moreover, the study found that altruism and prejudice significantly mediated the relationship between religiosity and health. Specifically, individuals with a higher level of religiosity held more altruistic attitudes toward other community members, society, and the world, which, in turn, improved their overall health. At the same time, those with a higher score on religiosity possessed less prejudice against socially disadvantaged and marginalized groups, which led to better overall health. In summary, the results of this study are consistent with our hypotheses of religiosity as a significant predictor of health and well-being through both direct and indirect mechanisms. These findings, along with the evidence from many other regions of the world showing a salutary relationship of religiosity to health, suggest that religiosity may promote individual health in diversified religious contexts.

## Figures and Tables

**Figure 1 ijerph-19-05605-f001:**
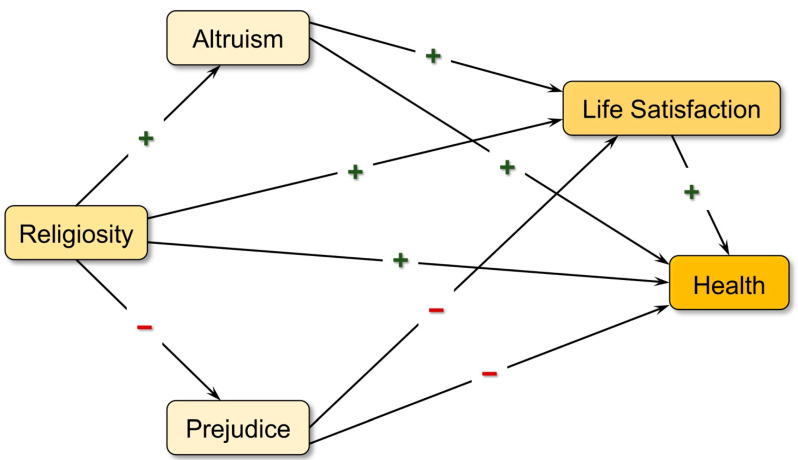
The Theoretical framework.

**Figure 2 ijerph-19-05605-f002:**
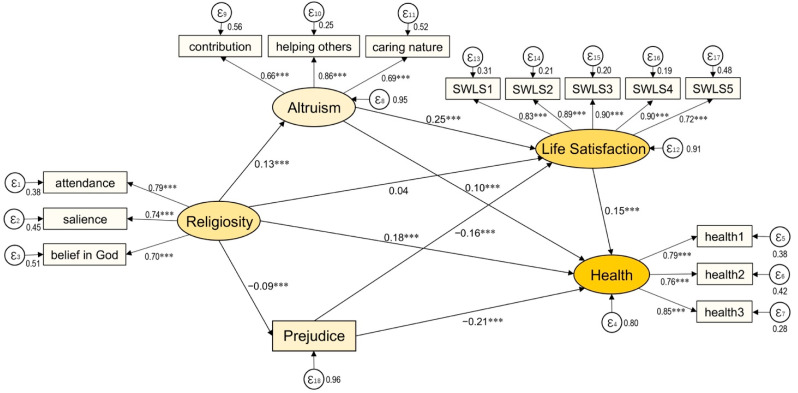
Chi-square (χ^2^) = 805.228, d.f. = 132; Root Mean Square Error of Approximation (RMSEA) = 0.040; Comparative Fit Index (CFI) = 0.973; Tucker–Lewis Index (TLI) = 0.963. All of the coefficients are standardized. The model controls for age, gender, birthplace, and education level. Note: *** *p* < 0.001, two-tailed.

**Table 1 ijerph-19-05605-t001:** Descriptive Analysis (*N* = 3258).

Variables				
**Demographic characteristic**	**%**	**S.D.**	**Min**	**Max**
Female	52.30%	0.50	0	1
Birthplace (Macau)	52.46%	0.50	0	1
**Demographic characteristic**	**Mean**	**S.D.**	**Min**	**Max**
Age (years)	43.78	17.11	16	96
Education level	2.93	1.69	0	7
**Problem gambling**	0.06	0.34	0	3
**Religiosity**				
Religious activity attendance	1.06	2.13	0	7
Religious salience	2.15	1.27	1	5
Belief in God	3.09	2.05	1	6
**Altruism**				
Social contribution	4.55	1.55	1	7
Helping others	4.87	1.45	1	7
Caring for nature	4.82	1.49	1	7
**Prejudice**	1.77	1.81	0	10
**Life satisfaction**				
Life is close to ideal	5.29	1.13	1	7
Conditions are excellent	5.03	1.26	1	7
Satisfied with life	5.04	1.25	1	7
Have important things	4.95	1.29	1	7
Would change almost nothing	4.68	1.51	1	7
**Health**				
Health condition compared to peer(s)	3.28	0.80	1	5
Health condition compared to last year	3.07	0.79	1	5
General health	2.69	0.96	1	5

**Table 2 ijerph-19-05605-t002:** Types of Religious Affiliation.

Religious Affiliation	N	%	Cumulative%
No religion	2228	68.38	68.38
Buddhism	506	15.53	83.91
Catholicism	212	6.51	90.42
Protestantism	154	4.73	95.15
Other	126	3.87	99.02
Daoism	22	0.68	99.70
Mazuism	5	0.15	99.85
Islam	5	0.15	100.00

**Table 3 ijerph-19-05605-t003:** The Decomposition of the Effects of Religiosity on Life Satisfaction and Health.

	Std. Coef.	S.E
**Religion on life satisfaction**		
Direct effect		
Religion →satisfaction	0.038	0.013
Indirect effect		
Religion → altruism → satisfaction	0.031 ***	0.007
Religion → prejudice → satisfaction	0.014 ***	0.004
Total effect	0.083 ***	0.014
**Religion on health**		
Direct effect		
Religion → health	0.181 ***	0.012
Indirect effect		
Religion → altruism → health	0.012 **	0.004
Religion → prejudice → health	0.018 ***	0.005
Religion → satisfaction → health	0.006	0.004
Religion → altruism → satisfaction → health	0.005 **	0.001
Religion → prejudice → satisfaction → health	0.002 **	0.001
Total effect	0.225 ***	0.012

Note: *** *p* < 0.001, ** *p* < 0.01, two-tailed.

## Data Availability

The data presented in this study are available on request from the corresponding author.

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
