# Peer review of "In the Shadow of the Casinos: The Relationship between Religion and Health in Macau"

_ijerph, 2022, doi:10.3390/ijerph19095605_

Round 1

Reviewer 1 Report

The research was very well conducted and data properly analyzed. 

Nevertheless, the relationship between religiosity and health was found to have important mediator variables such as altruism and prejudice.

The discussion show that other circumstances religiosity may led to different impact on altruism and prejudice and, then, different influence on health and wellbeing. That's a very important issue that should be raised in the abstract.

1 Abstract:

As the measurement scale is based on health perception the abstract should address this approach changing line 10 to 12 from "This study tried 10 to address this gap by developing a theoretical model to examine how religiosity is related to life satisfaction and health in a non-Western culture." To "This study tried 10 to address this gap by developing a theoretical model to examine how religiosity is related to life satisfaction and health 'perception' in a non-Western culture. "

 2 Altruism and religious activities line 103:

The reference cited 65 says the opposite of the statement, it analises (page 167)  the altruistic hypocrisy where  "intrinsic and orthodox religion foster compassionate and caring beliefs, yet the likelihood of actually helping other decreases as those religious orientations increase".

 3 Expectation on altruism in Macao:

Lines 117 to 119 propose a relationship between Macao residents being less altruistic due to gambling environment without reference or evidence.

 4 Altruism as a mediating variable:

Why don't use "altruism" as a causal variable instead of mediating?

 5 Perception on health and the confirmation bias:

The authors could discuss the possibility of confirmation bias affect results.

Author Response

Dear reviewer,

Kind regards.

Reviewer 2 Report

The title properly reflects the subject of the manuscript. The abstract provides an accessible summary of the article. The keywords accurately reflect the content. The introduction explains clearly what is the main question addressed by the research. The text is clear and easy to read. The methods are used appropriate. The publication would make a useful contribution to the understanding of the the relationship between religiosity and overall health. However, the concluding part of the manuscript could be communicated better. It should demonstrate clearly what does this paper add to the subject area compared with other published material. The conclusions should reflect upon the aims - whether they were achieved or not? According to me, the conclusions should be based on the evidence and arguments presented. The key messages should be evidence-based, i.e. they should clearly describe what the data show.

Author Response

Dear reviewer,

Kind regards.

Reviewer 3 Report

This work focused on very interesting issues and I think it presents three main strengths:

  1. The attempts to explain how the relation between religion and health can be articulated;
  2. The sample procedure and the sample dimension;
  3. The investigation of a rather unknown context.

However, I believe the work should be strengthened in line with the following suggestions.

In the literature recognition about the role of religiosity as a protective factor for one’s overall health, some additional references can be added, such as:

  1. In psychosocial literature about religiosity, the definition of religions as “systems of meanings” (involving beliefs, values and a subjective feeling of significance in life) can better explain these relations;
  2. A specification about the role of faith across lifespan, especially in elderly persons (e.g. Manuti et al., 2006, “Me, myself and God”) can be added.

You present the Macau scenario and it is a central point in the work (both “casino” and “Macau” are part of the title). In addition, you present Macau as “the largest commercial gambling market in the world”. Nonetheless, you did not make use of some specific variables concerning this, even as a control variable.

Why didn’t you choose to insert some specific variables (e.g. the gambling attitude or something like this)?

You talked about “deviant behaviours”, but it can fall into the social desirability effect, so how did you question it? The risk is considering Macau as a mere “context” where gathering data rather then as a “socio-cultural scenario”.

I think that a better connection between the section 1 and 2 should be created, e.g. through a premise before 2.1.

When you talk about “Altruism”, studies in religious population were presented. Are there studies concerning altruism in atheistic/agnostic population?  

As for the measure of “Religiosity”, why did you not use the same point scale (rather than 7, 5 and 6 point)? In addition, in the theoretical background you present the difference between intrinsic/extrinsic religiosity. Did you think it could be useful to better explain your results?

In psychosocial literature, the measures of altruism are really numerous. Why did you choose this one, including caring for nature? More generally, all the selected variables can be assessed through a very big variety of measures. With the exception of life satisfaction, which used a “strong” instrument, I think you should better explain how the measures were selected. Once again, the measure of “health” is founded on three items (a general one and two comparative ones). Why did you choose these measures rather than a more articulated one (e.g. a measure of physical and mental health)?

Looking at the sample, rather 47% were not born in Macau. How can you explain this data?

In table 2 “Types of Religious Affiliation”, 68.39% was “no religious”. Did you insert both atheist and agnostic in the socio-demographic section of the interview or do you simply put “not religious”? This difference could be interesting in line with the so wide number.

Author Response

Dear reviewer,

Kind regards.

Reviewer 4 Report

Impressive study! The differences regarding prejudice (ll. 437-450) are worth further study.

Author Response

Dear reviewer,

Kind regards.

Round 2

Reviewer 3 Report

Thank you for your comprehensive answers and integrations, I believe the paper now is widely strengthened

Author Response

Dear reviewer,

We are very appreciative of these insightful comments you provided on our manuscript. We have checked our spelling and grammatical errors.

Yours sincerely.